# The Effect of Pre-Analytical Conditions on Blood Metabolomics in Epidemiological Studies

**DOI:** 10.3390/metabo9040064

**Published:** 2019-04-03

**Authors:** Diana L. Santos Ferreira, Hannah J. Maple, Matt Goodwin, Judith S. Brand, Vikki Yip, Josine L. Min, Alix Groom, Debbie A. Lawlor, Susan Ring

**Affiliations:** 1Medical Research Council Integrative Epidemiology Unit at the University of Bristol, Bristol BS8 2BN, UK; hannah.maple@bio-techne.com (H.J.M.); matt.goodwin@bristol.ac.uk (M.G.); judith.brand@bristol.ac.uk (J.S.B.); josine.min@bristol.ac.uk (J.L.M.); alix.groom@bristol.ac.uk (A.G.); d.a.lawlor@bristol.ac.uk (D.A.L.); s.m.ring@bristol.ac.uk (S.R.); 2Population Health Sciences, Bristol Medical School, University of Bristol, Bristol BS8 2PS, UK; vikki.yip@bristol.ac.uk; 3Clinical Epidemiology and Biostatistics, School of Medical Sciences, Örebro University, 701 85 Örebro, Sweden; 4Bristol National Institute of Health Research Biomedical Research Centre, Bristol BS1 3NU, UK

**Keywords:** metabolomics, serum, plasma, nuclear magnetic resonance, pre-analytical phase

## Abstract

Serum and plasma are commonly used in metabolomic-epidemiology studies. Their metabolome is susceptible to differences in pre-analytical conditions and the impact of this is unclear. Participant-matched EDTA-plasma and serum samples were collected from 37 non-fasting volunteers and profiled using a targeted nuclear magnetic resonance (NMR) metabolomics platform (*n* = 151 traits). Correlations and differences in mean of metabolite concentrations were compared between reference (pre-storage: 4 °C, 1.5 h; post-storage: no buffer addition delay or NMR analysis delay) and four pre-storage blood processing conditions, where samples were incubated at (i) 4 °C, 24 h; (ii) 4 °C, 48 h; (iii) 21 °C, 24 h; and (iv) 21 °C, 48 h, before centrifugation; and two post-storage sample processing conditions in which samples thawed overnight (i) then left for 24 h before addition of sodium buffer followed by immediate NMR analysis; and (ii) addition of sodium buffer, then left for 24 h before NMR profiling. We used multilevel linear regression models and Spearman’s rank correlation coefficients to analyse the data. Most metabolic traits had high rank correlation and minimal differences in mean concentrations between samples subjected to reference and the different conditions tested, that may commonly occur in studies. However, glycolysis metabolites, histidine, acetate and diacylglycerol concentrations may be compromised and this could bias results in association/causal analyses.

## 1. Introduction

The development of high-throughput methods for quantifying multiple ‘omic traits in large-scale epidemiological studies, using stored biosamples, has the potential to rapidly advance our understanding of how human physiology and metabolism vary across the lifecourse. Applying these methods to existing samples from very densely pheno- and genotyped cohorts also has the potential to enhance our understanding of causal mechanisms for disease. To do this in a scientifically rigorous way, further information is required regarding the potential impact of differing pre-analytical conditions (e.g., variations in incubation duration and temperature before centrifugation) of stored samples that are likely to vary between cohorts and potentially within the same cohort over time. For example, since sample collection from longitudinal epidemiological cohorts are collected over many decades, it may be impossible to ensure that identical protocols are used throughout the lifetime of the study. In clinical cohorts and in pregnancy/birth cohorts, such as the Avon Longitudinal Study of Parents and Children [1,2] and Born in Bradford [3] cohorts, some samples (e.g., during antenatal care or for cord-blood) are obtained during routine clinical practice where health care needs takes precedence over speedy sample processing and rapid storage. Yet these samples, collected early in life, are often the most valuable for research related to the long-term effects of childhood exposures. Other cohorts face different challenges, for example, the Health Survey for England [4] and Understanding Society [5] collect biological samples in participants’ homes, which is likely to increase pre-analytical variability.

Additionally, we are increasingly interested in cohort collaborations (e.g., University College London-London School of Hygiene & Tropical Medicine-Edinburgh-Bristol [6] consortium, COnsortium of METabolomics Studies [7], Cohorts for Heart and Aging Research in Genomic Epidemiology [8] consortium) and broader international comparisons, for example, between determinants of health and well-being in low-, middle- and high-income countries and how these differ. Consequently, we often need to combine data from samples collected in very different circumstances. For all these reasons, it is important to recognise that biosample collection protocols are often a compromise between reducing any effects of pre-analytical variation with maximising the amount of data that can be obtained from samples, within the financial and practical restraints of a given collection sweep.

In clinical chemistry, pre-analytics account for 60–80% of laboratory errors [9,10,11,12], and current standard operating procedures for blood handling in metabolomics are generally based on best practices [13] (not evidence-based) for conventional biochemistry tests [9,14]. While several studies have explored the impact of pre-analytical conditions on a small number of commonly assessed biomarkers in epidemiology [13,15,16,17,18], metabolomics—the simultaneous quantification of large numbers of metabolic traits—has particular challenges as different metabolites may have different susceptibilities to degradation [19,20,21,22,23].

The aim of this study was to determine the differences in metabolite concentrations between reference and six pre-analytical conditions, which reflect conditions potentially arising during sample collection and processing in large-scale epidemiological studies. We used both Spearman’s rank correlation and differences in mean concentrations to address this aim. Rank correlations are useful for considering any impact of pre-analytical effects on subsequent association or causal effect analyses using metabolomics data, as bias of these results is likely to be minimal if participants are still ranked similary with respect to metabolite concentrations. Differences in mean concentrations are important when describing distributions of a metabolite in a population or comparing differences between populations.

Throughout this manuscript, we use the term pre-storage instead of (sample) pre-processing to indicate the phase of the metabolomic workflow that occurs before long-term storage of samples. This was done to avoid confusion with the term (data) pre-processing which is used, particularly in untargeted metabolomic studies, to mean the several steps undertaken after metabolomics data acquisition (e.g., peak alignment, normalization).

## 2. Results

Characteristics of study participants are shown in Table 1, with metabolic trait distributions in Appendix A. Most participants were female (78%) and 46% were between 21 and 35 years old; 41% had drunk alcohol within the 24 h period prior to blood sampling.

The following main conclusions can be drawn from our analyses of the effect of pre-analytical conditions on blood metabolic traits:72% and 93% of metabolic traits across pre- and post-storage conditions, respectively, had Spearman’s correlations, between metabolite concentrations of samples subjected to reference and variant conditions, above 0.8 (Appendix A; Figure 1, Figure 2, Figure 3, Figure 4, Figure 5, Figure 6, Figure 7 and Figure 8; Appendix A). Exceptions are described in the next two bullet points.In both biofluids, diacylglycerol and histidine concentrations had, generally, Spearman’s correlation coefficient of ~0.5 or lower between reference and variant pre- and post-storage conditions (Figure 1, Figure 2, Figure 3, Figure 4, Figure 5, Figure 6 and Figure 7; Appendix A).Sphingomyelins, conjugated linoleic acid, lactate, glucose (serum), pyruvate (serum), glycerol (serum), phenylalanine (EDTA-plasma) and acetate had Spearman rank correlations between ~0.7 and −0.06 between concentration of samples subjected to variant pre-storage conditions and reference.Glycolysis-related metabolites (glucose, lactate, pyruvate) and amino acids showed marked differences in mean concentration between reference and pre-storage conditions and post-storage conditions (amino acids only) (Appendix A).Branched and aromatic amino acids had smaller changes in mean differences from the reference and variant pre-storage conditions in EDTA-plasma than in serum (Appendix A).For pyruvate, pre-storage delay and temperature appeared to interact, with mean levels slightly decreasing/stable from reference levels per 24 h at 4 °C and increasing per 24 h at 21 °C (Appendix A).As an illustration of the magnitude of effects in serum, mean levels of glucose decreased by 0.91 mmol/L (95% CI: 0.87 to 0.95) and 1.9 mmol/L (95% CI: 1.7 to 2.1) for each 24 h delay, compared with the reference sample, at 4 °C and 21 °C, respectively. Pyruvate levels decreased by 0.013 mmol/L per 24 h (95% CI: 0.008 to 0.018) at 4 °C but increased by 0.73 mmol/L per 24 h (95% CI: 0.64, 0.82) at 21 °C.Post-storage conditions affected histidine, phenylalanine and low-density lipoprotein (LDL) particle size, with differences in mean levels of up to 1.4 SD from reference (Appendix A).

## 3. Discussion

In this experiment, we showed that most metabolites, including lipids, lipoproteins and fatty acids, are minimally affected by different sample pre-analytical conditions that were designed to reflect plausible variation in large-scale epidemiological studies that collected samples at different time points and different populations. We explored the rank correlation and differences in mean levels of different pre-analytical conditions for over 151 metabolic traits. The pre-analytical conditions that we explored reflect realistic situations in large-scale epidemiological studies. We believe this is an important contribution to the literature given the increasing use of high-throughput metabolomics in stored samples from large epidemiological studies.

In this study, 72% and 93% of metabolite traits across pre- and post-storage conditions, respectively, had Spearman’s correlations, between metabolite concentrations of samples subjected to reference and variant conditions above 0.8. This suggests that, for most measurements, there is little effect of different pre-analytical conditions on metabolic concentrations or, where there are effects, these are similar across paticipants. Most metabolites also had similar mean concentrations between the reference and different tested pre-analytical conditions (91% and 97%, across pre and post-storage conditions, respectively, had <0.5 SD), with glycolysis and some amino acids in particular having both, low rank correlations and high differences in mean concentrations (up to 1.4 SD).

Previous studies examining pre-analytical effects on metabolites have largely explored differences in mean levels and measured this using clinical chemistry tests [15,16,17,18,24], mass spectrometry [14,25,26,27] and NMR [19,24,28,29,30,31,32], and with targeted and untargeted platforms. Where it is possible to make comparisons of these previous studies with our results, these are summarized in Appendix A. Pre-storage delay and incubation temperature of uncentrifuged samples can cause changes to serum and plasma metabolomes because blood cells and enzymes are still metabolically active inside the sample tube, resulting in the uptake and release of metabolites. Concordant findings include the reported stability of triglycerides [16,17,18], high density lipoprotein-cholesterol (C) [18], low density lipoprotein-C (4 °C) [18], total cholesterol (4 °C) [18], apolipoprotein A-I and B [18] and creatinine [18]. Our observed changes in glucose, pyruvate, lactate and alanine agree with previous studies [17,18,19,26,29], and are likely due to blood cell activity, primarily red blood cells (RBC) [23,33]. The absence of RBC in post-storage samples explains why these metabolites are minimally affected by the tested post-storage conditions [17,20]. Increases in concentrations of most amino acids with the pre- and post-storage (non-ideal) conditions that we tested here are also consistent with previous studies [14,29]. One possible reason for this is protein degradation, for example, in the case of phenylalanine [14,29] accumulation, as it is an uncommon component of proteins and has few degradation pathways [14]. Glutamine decreased [14,29], converted into glutamate, and high levels of the latter prevented reliable quantification of pyruvate, since pyruvate only gives one peak in the ^1^H NMR spectrum, which can be overlapped by glutamate signals. Our results also show that amino acids in serum are more affected by pre-storage conditions than in EDTA-plasma, possibly due to the inhibition of metal-dependent proteases by EDTA in the latter, and the activation of a range of proteases during coagulation, in the former. Histidine and acetate had opposite changes in serum and plasma, likely due to the influence of the presence/absence of anti-coagulant. Post-storage effects were less pronounced than pre-storage ones, indicating that most of the sample degradation is related to blood cell activities [23,34]. Nevertheless, we found changes in histidine and phenylalanine and, although not directly comparable to the conditions we tested, others [14,19] have reported changes in some metabolites when delays occur after centrifugation. These are potentially caused by enzymes released due to cell damage during centrifugation and other proteins [14,19,34]. While our study has explored the effects on metabolite concentrations using one quantitative NMR platform, we would anticipate similar results with other metabolomic platforms that measure the same metabolites, as pre-analytic conditions affect specific metabolites (as described above) rather than being related to the type of metabolomic platform (only concentration accuracy and precision may differ across platforms).

### Potential Bias in Epidemiological Studies of Pre-Analytic Conditions

Rank correlations are useful for considering any impact of pre-analytical effects on subsequent association or causal effect analyses using metabolomics data (i.e., analytical studies), because the bias of these results is likely to be minimal if participants are still ranked similarly with respect to metabolite concentrations. Differences in mean concentrations are important in descriptive studies where the aim is to describe the distribution of a metabolite in a population or between two populations. For example, our results suggest that, on average, glucose levels from samples that were left for 24 h at 21 °C before being processed and then placed in long-term storage would be 1.9 mmol/L lower than the true value. Therefore, if mean levels of glucose from one study (e.g., from a population of South Asian adults) where samples had been left for 24 h at 21 °C were compared to those from another study (e.g., from a population of European adults) where samples had been left for no longer than 1.5 h at 4 °C (our reference/ideal conditions), we may incorrectly conclude that mean glucose concentrations are lower in South Asian compared to European adults or mask higher concentrations in South Asians. In our study, there was some consistency between the Spearman rank correlation and difference in mean concentration results. In particular, pre-analytical conditions for glycolysis metabolites and some amino acids affected both rank correlation and mean differences, and studies with less than ideal pre-analytical conditions should consider whether any analyses might be biased for these metabolites.

In analytical studies, the extent of any bias or random errors will depend on the particular research question, the exposure, the outcome and covariable measurements included in the analyses, as well as sample size. In addition, in association and causal effect analyses, random error and systematic bias must be considered relative to exposures, outcomes and covariables. For example, if we were interested in the potential effect of multiple metabolites (exposures) on incident coronary heart disease (CHD) (outcome), systematic bias would occur if the effect of pre-analytic conditions on metabolites differed between those who experienced CHD and those who did not (i.e., pre-analytical conditions would be associated with both, metabolite concentrations and incidence of CHD, therefore pre-analytical conditions would be a confounder of the association between metabolites and CHD). This could occur if samples from cases of CHD are handled more often (e.g., have more freeze–thaw cycles) than those of controls, which may occur in prospective nested case control studies. These cycles may enhance the impact of pre-analytical effects, particularly for labile metabolites. However, we acknowledge that we have not assessed that matter here. Random error could also bias results in this example. Since researchers at the time of collecting and processing serum/plasma are unlikely to know the future risk of CHD and as sample processing differences are likely to affect all samples similarly within a study, this may be more likely than systematic bias. In this case, effects of the pre-analytical conditions are random with respect to CHD and the expectation with random error is that it biases results towards the null. Let us consider a second example where metabolites are outcomes, for example, if we were interested in the effect of body mass index (BMI) (exposure) on metabolites (outcomes). As in the first example, if pre-analytical conditions that result in measurement error in the metabolites are associated with BMI, there will be systematic bias that, depending on the extent of measurement error and how strongly, and in which direction, it is related to BMI and confounders, could bias results towards or away from the null to a large or small extent. If the pre-analytical conditions are not related to BMI, the expectation would be bias towards the null. A key thing here is whether BMI (or other exposure) was measured some time before bio-samples were collected (i.e., in prospective studies) or at the same time (cross-sectional studies). It is not plausible that pre-analytical conditions could (retrospectively) influence prior measured BMI, and therefore in prospective studies there would be random error with an expectation of bias towards the null. However, in cross-sectional studies, and in particular if the exposure of interest was another biomarker measured in the same samples (e.g., if we were looking at the association of a cytokine with metabolites), then systematic bias is plausible. Regardless of metabolites being exposure or outcome in the analysis, associations between pre-analytical conditions and covariables also need to be considered. As with the examples above, random error would be expected to bias metabolite–covariable associations towards the null and this may result in residual confounding in the main analyses.

The above discussion and examples refer to single studies with single centres, whereas metabolomic (and other ‘omic analytical studies) are increasingly multi-studies, something that is important for replication and increasing sample size. Systematic bias could occur in multi-centre studies or collaboration pooling results from a number of different studies, if sample handling and the likelihood of having an outcome (e.g., CHD) or the distribution of an exposure (e.g., BMI) varies by centre.

In summary, for most of the metabolites that we assessed there were high rank correlations and low differences in means between plausible, but less than ideal pre-analytical conditions, compared with the ideal. However, for the metabolites that show low rank correlation and/or high differences in mean levels (specifically glycolysis metabolites, histidine, acetate and diacylglycerol), researchers should consider the extent to which these effects might have biased results towards the null (due to random error or systematic bias that masks a true effect) or away from the null (due to systematic bias or the introduction of residual confounding). Ideally, one would want to compare results with other studies in similar populations that had “ideal” pre-analytic conditions. Such bias and confounding are likely to be more pronounced in studies with small sample sizes. In large collaborations where results from several studies are pooled, we would recommend specifically exploring the extent to which pre-analytic processing might have influenced between-study heterogeneity.

## 4. Materials and Methods

Samples from 37 healthy volunteers were collected. Exclusion criteria for the study were the following: clotting/bleeding disorders, anaemia, use of anti-coagulant medication or insulin treatment and presence of blood borne viruses. Participants provided written informed consent and completed a questionnaire. Ethical approval for the study was obtained from the South West Frenchay Proportionate Review Committee, Bristol, UK, reference 14/SW/0087. An overview of the experimental design is given in Figure 9. Ten venous blood tubes were drawn for each non-fasting participant (5 EDTA-plasma, 5 serum). Further details are provided in Appendix B.

### 4.1. Reference Samples

All samples were processed within 1.5 h of blood withdrawal. Once centrifuged, serum and EDTA-plasma samples were aliquoted into 1.5 mL microtubes (E1415-2240, STARLAB, Milton Keynes, UK) and immediately frozen and stored at −80 °C for one month prior to NMR analysis. Prior to profiling, frozen samples were thawed in the refrigerator (4 °C) overnight, prepared with sodium phosphate buffer [35,36,37] and then immediately run through the NMR spectrometer.

### 4.2. Pre-Storage Handling: Effect of Pre-Centrifugation Delay and Temperature

We compared metabolic trait concentrations, quantified by the NMR platform, between four pre- and two post-storage conditions and the reference samples. The four pre-storage conditions had the following combination of incubation temperature/duration before centrifugation (i) 4 °C for 24 h; (ii) 4 °C for 48 h; (iii) 21 °C for 24 h; and (iv) 21 °C for 48 h. These conditions were chosen to reflect pre-storage variations likely to occur in epidemiological studies. Participant-matched serum and EDTA-plasma samples were processed according to these four conditions.

### 4.3. Post-Storage Handling: Effect of Buffer Addition Delay and NMR Profiling Delay

Two variations to the standard Nightingale Health^©^ NMR protocol [35,36,37] were investigated to evaluate the impact of sample preparation and instrumental analysis delay. Matched serum and EDTA-plasma aliquots of each participant previously prepared according to the reference pre-storage conditions, described above, were subjected to two post-storage conditions. Samples thawed overnight (4 °C) and afterwards were either (i) left for 24 h, at 4 °C in the dark, before addition of sodium phosphate buffer followed by immediate NMR analysis (buffer addition delay); or (ii) addition of buffer before being left for 24 h, at 4 °C in the dark, then NMR profiling (NMR analysis delay).

### 4.4. Nuclear Magnetic Resonance Metabolomics Platform

A high-throughput NMR metabolomics platform, Nightingale Health^©^, at the University of Bristol, was used to quantify up to 151 lipoproteins, lipids and metabolites in serum and plasma. The platform applies a single experimental setup, providing the simultaneous quantification of routine lipids, 14 lipoprotein subclasses and lipids transported by these particles, various fatty acids (FA) and FA traits (e.g., chain length, degree of unsaturation), amino acids, ketone bodies, glycolysis and gluconeogenesis-related metabolites, fluid balance and one inflammation metabolite. Most of these are quantified in clinically meaningful concentrations (e.g., mmol/L), and particle size in nm. Details of this platform and its use in epidemiological studies have been described elsewhere [38,39,40,41]. Pyruvate, glycerol and glycine are not quantified in EDTA-plasma samples due to the interfering resonances of EDTA on their signals.

### 4.5. Statistical Analysis

We examined both the Spearman’s rank correlation and differences in mean concentrations of metabolic trait values/concentrations, between reference and six pre-analytical conditions likely to occur in large-scale epidemiological studies.

All analyses were conducted using participant-matched serum and EDTA-plasma samples. As bias in subsequent association or causal analyses involving metabolic traits is likely to be minimal if participants are still ranked similary with respect to metabolite levels (even if there are differences in mean levels), we considered the rank correlation coefficients between the reference and each pre-analytical conditions to be our main analyses. Confidence intervals for Spearman’s rank correlation coefficients were computed by bootstrap with 200 repetitions.

Differences in mean concentrations of metabolite levels between reference and each pre-analytic condition are particularly important in analyses of distributions within and between populations (i.e., descriptive studies). To assess changes in mean concentrations of metabolic traits due to the different pre-analytical conditions, we used linear regression models with random intercepts. Such models account for within individual clustering of observations as each individual participant contributes with multiple samples for analysis.

All metabolic traits were scaled to standard deviation (SD) units (by subtracting the mean and dividing by the standard deviation); this was done separately for EDTA-plasma and serum, and pre- and post-storage conditions. Standardization allows the comparison of metabolic traits with different units and/or different concentration ranges. Results in measured concentration units (e.g., mmol/L) are provided in the Appendix A. Full specification of the models can be found in Appendix B. First, we evaluated the impact of pre-storage duration, before centrifugation, by keeping temperature constant (at 4 °C and 21 °C). Incubation duration was entered as a continuous term (levels: 0 = reference, 1 = 24 h, 2 = 48 h) in the model with betas representing the standardized mean difference in metabolite concentration per 24 h increment in incubation time at 4 °C and 21 °C, respectively. Next, we investigated the impact of post-storage conditions to estimate the standardized mean difference in metabolite concentration comparing delays in buffer addition and NMR profiling to the reference (levels: 0 = reference, 1 = buffer addition delay or NMR profiling delay).

Metabolic traits are highly correlated [38]. Nine principal components, in this study, explain at least 95% of the variance of serum and EDTA-plasma metabolome (see Appendix A), and this number is used to correct conventional *p*-value thresholds for nominal significance using the Bonferroni method (alpha = 0.05/9 = 0.006) [41,42]. Here, we focused on effect size and precision [43,44]. The rationale for defining the number of independent tests via principal component analysis (PCA) has been discussed previously [25,29,30], and more information is available in Appendix A.

In addition, to better appreciate the effects of pre-storage temperature and duration of incubation on the overall metabolic profile [45], we also used PCA [46]. In PCA scores plots, the metabolic profile of each sample is shown as a single data point, therefore each participant is represented by five data points corresponding to each pre-storage condition. Samples (data points) close to one another have similar metabolic composition in comparison to samples further apart [47]. Therefore, if pre-storage conditions do not impact metabolic profiles, samples will not cluster by pre-storage conditions.

Statistical analyses were conducted using R version 3.0.1 (R Foundation for Statistical Computing, Vienna, Austria).

## 5. Conclusions

Most serum and EDTA-plasma metabolic traits, quantified by Nightingale Health^©^ NMR platform (87% of measured traits are lipid-related), are minimally affected by the pre- and post-storage conditions tested, with the exception of glycolysis metabolites, histidine, acetate and diacylglycerol. Potential bias, and its magnitude and direction, of association and causal effect estimates as a result of the impact of pre-analytical conditions, on some metabolites, will depend on the specific research questions being addressed. This should be considered for analyses with glycolsis metabolites, histadine, acetate and diaglycerol in particular. Furthermore, in large collaborations where data are pooled from different studies, the possible impact of pre-analytical conditions on between-study heterogeneity should be explored.

## Figures and Tables

**Figure 1 metabolites-09-00064-f001:**
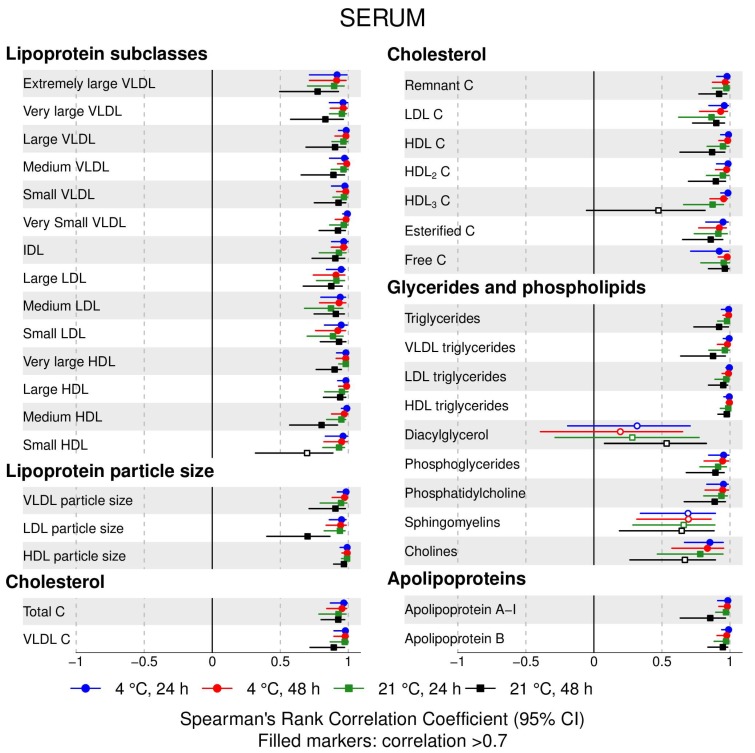
Serum, pre-storage handling effects: Spearman’s rank correlation coefficients between metabolic trait concentration/value in reference samples (4 °C, 1.5 h) and samples incubated at (i) 4 °C, 24 h; (ii) 4 °C, 48 h; (iii) 21 °C, 24 h; and (iv) 21 °C, 48 h, before centrifugation (correlations for lipoprotein trait details are given in Appendix A). Spearman’s rank correlation coefficients and 95% confidence intervals (CIs) are listed in Appendix A. C, cholesterol; IDL, intermediate-density lipoprotein; LDL, low-density lipoprotein; HDL, high-density lipoprotein; VLDL, very low-density lipoprotein.

**Figure 2 metabolites-09-00064-f002:**
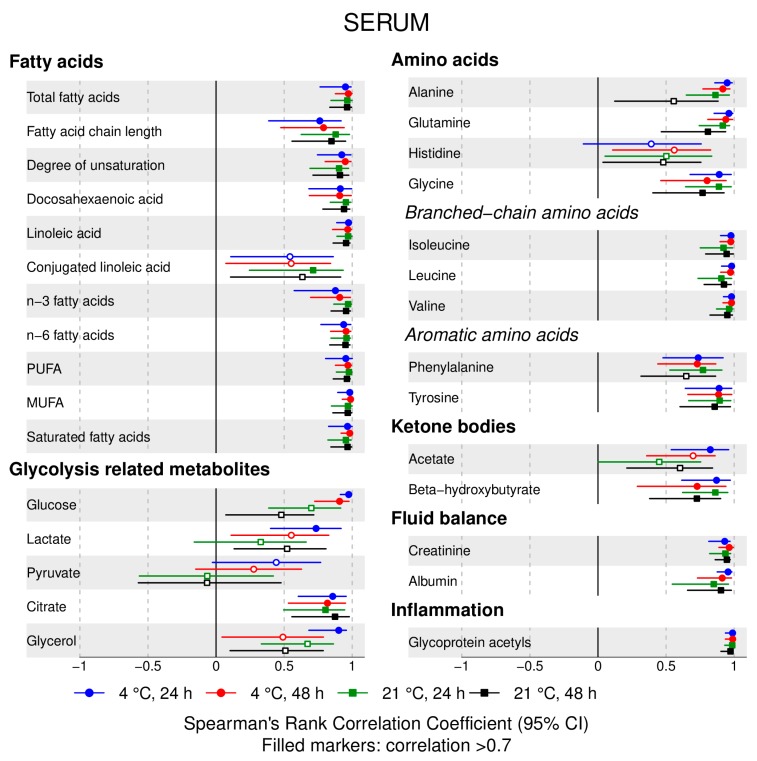
Serum, pre-storage handling effects (Figure 1 continued): Spearman’s rank correlation coefficients between metabolic trait concentration/value in reference samples (4 °C, 1.5 h) and samples incubated at (i) 4 °C, 24 h; (ii) 4 °C, 48 h; (iii) 21 °C, 24 h; and (iv) 21 °C, 48 h, before centrifugation (correlations for lipoprotein trait details are given in Appendix A). Spearman’s rank correlation coefficients and 95% confidence intervals (CIs) are listed in Appendix A. MUFA, monounsaturated fatty acids; PUFA, polyunsaturated fatty acids.

**Figure 3 metabolites-09-00064-f003:**
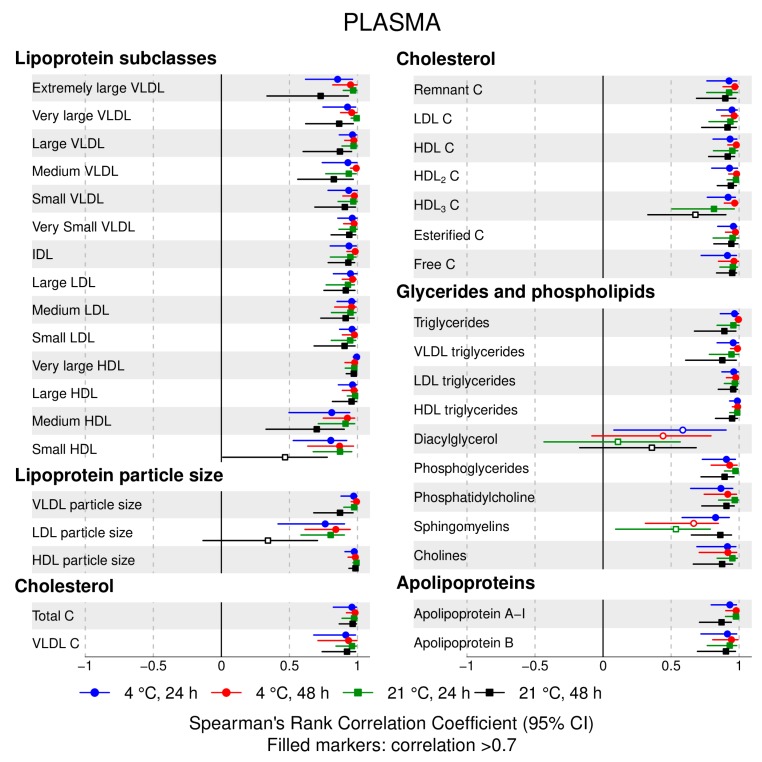
EDTA-plasma, pre-storage handling effects: Spearman’s rank correlation coefficients between metabolic trait concentration/value in reference samples (4 °C, 1.5 h) and samples incubated at (i) 4 °C, 24 h; (ii) 4 °C, 48 h; (iii) 21 °C, 24 h; and (iv) 21 °C, 48h, before centrifugation (correlations for lipoprotein trait details are given in Appendix A). Spearman’s rank correlation coefficients and 95% confidence intervals (CIs) are listed in Appendix A. C, cholesterol; IDL, intermediate-density lipoprotein; LDL, low-density lipoprotein; HDL, high-density lipoprotein; VLDL, very low-density lipoprotein.

**Figure 4 metabolites-09-00064-f004:**
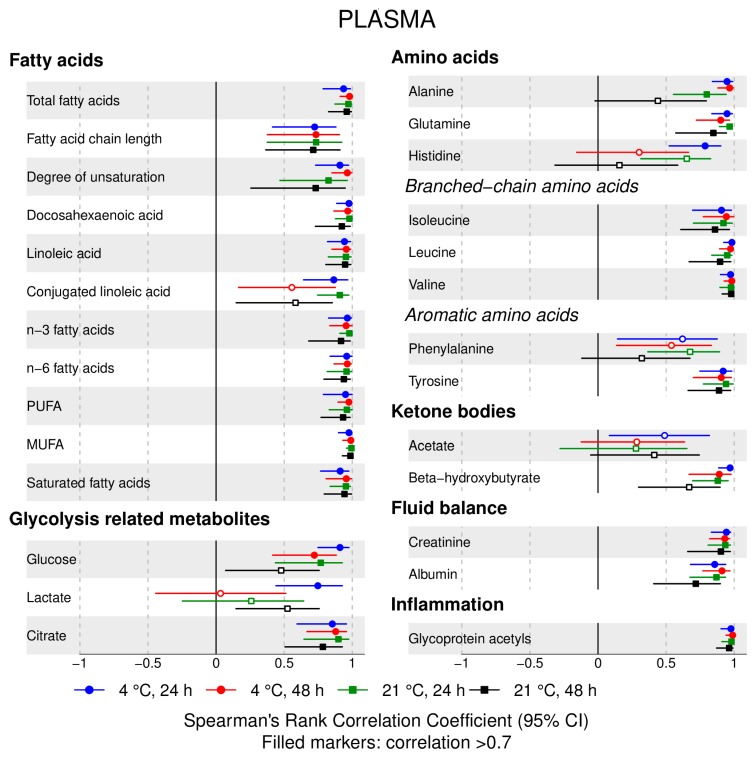
EDTA-plasma, pre-storage handling effects (Figure 3 continued): Spearman’s rank correlation coefficients between metabolic trait concentration/value in reference samples (4 °C, 1.5h) and samples incubated at (i) 4 °C, 24 h; (ii) 4 °C, 48 h; (iii) 21 °C, 24 h; and (iv) 21 °C, 48 h, before centrifugation (correlations for lipoprotein trait details are given in Appendix A). Spearman’s rank correlation coefficients and 95% confidence intervals (CIs) are listed in Appendix A. MUFA, monounsaturated fatty acids; PUFA, polyunsaturated fatty acids.

**Figure 5 metabolites-09-00064-f005:**
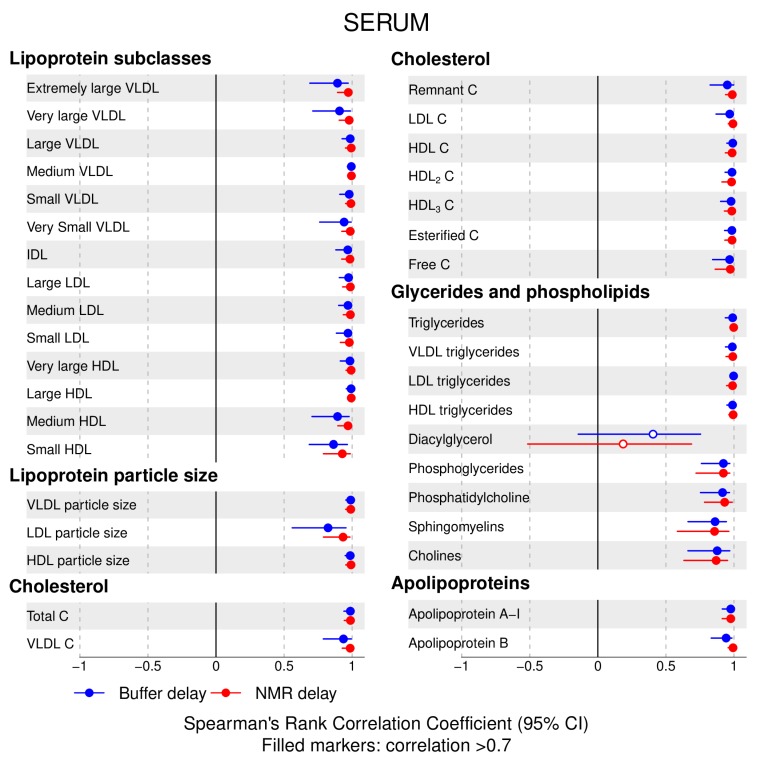
Serum, post-storage handling effects: Spearman’s rank correlation coefficients between metabolic trait concentration/value in reference samples (no buffer addition or nuclear magnetic resonance (NMR) analysis delays) and samples subjected to two variant post-storage conditions in which samples thawed overnight and afterwards (i) were left for 24 h before addition of sodium buffer followed by immediate NMR analysis (buffer delay); and (ii) addition of sodium buffer, then left for 24 h before profiling (NMR delay) (correlations for lipoprotein trait details are given in Appendix A). Spearman’s rank correlation coefficients and 95% confidence intervals (CIs) are listed in Appendix A. C, cholesterol; IDL, intermediate-density lipoprotein; LDL, low-density lipoprotein; HDL, high-density lipoprotein; VLDL, very low-density lipoprotein.

**Figure 6 metabolites-09-00064-f006:**
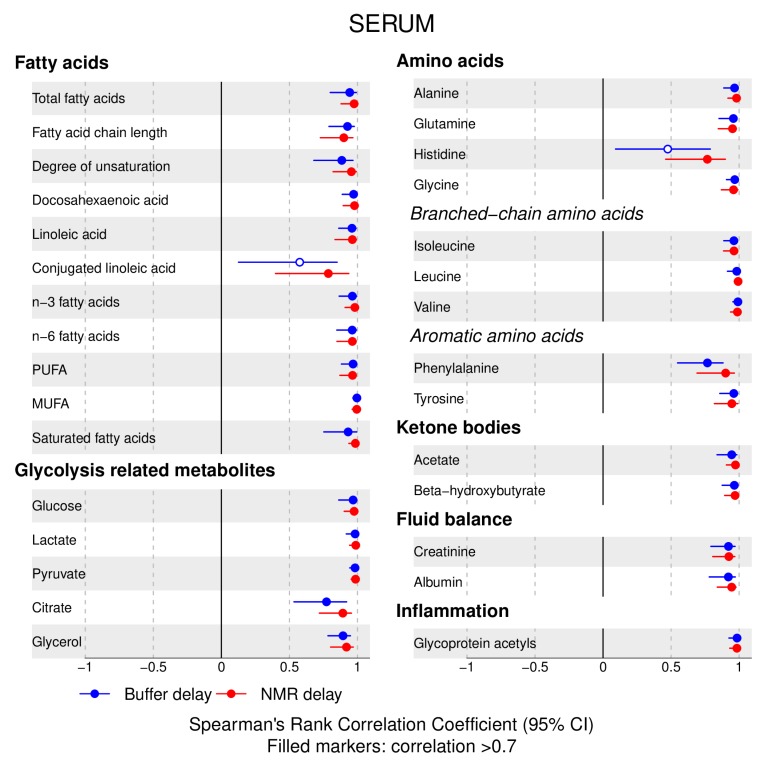
Serum, post-storage handling effects (Figure 5 continued): Spearman’s rank correlation coefficients between metabolic trait concentration/value in reference samples (no sample preparation or nuclear magnetic resonance (NMR) analysis delays) and samples subjected to two variant post-storage conditions in which samples thawed overnight and afterwards (i) were left for 24 h before addition of sodium buffer followed by immediate NMR analysis (buffer delay); and (ii) addition of sodium buffer, then left for 24 h before profiling (NMR delay) (correlations for lipoprotein trait details are given in Appendix A). Spearman’s rank correlation coefficients and 95% confidence intervals (CIs) are listed in Appendix A. MUFA, monounsaturated fatty acids; PUFA, polyunsaturated fatty acids.

**Figure 7 metabolites-09-00064-f007:**
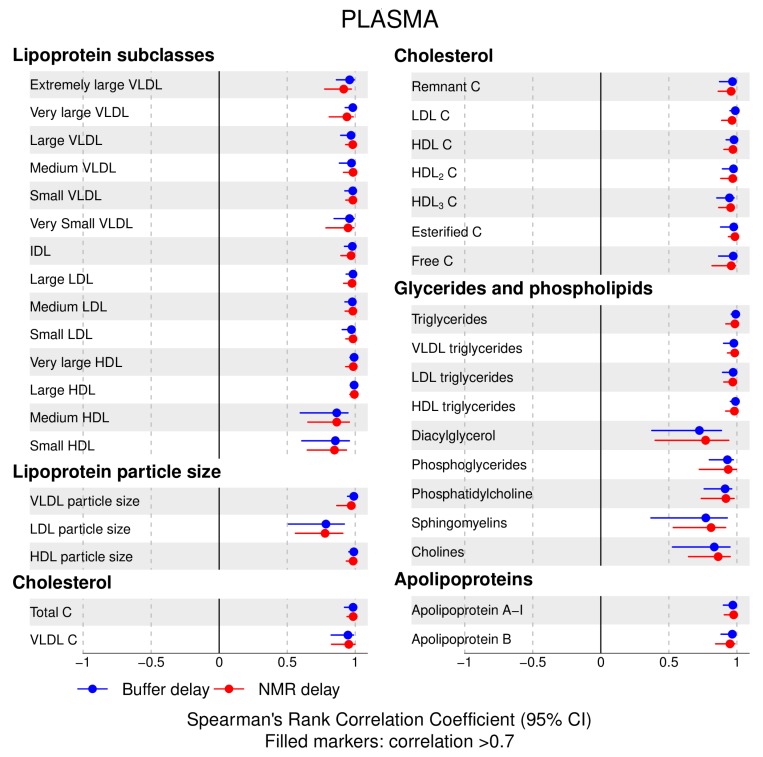
EDTA-plasma, post-storage handling effects: Spearman’s rank correlation coefficients between metabolic trait concentration/value in reference samples (no sample preparation or nuclear magnetic resonance (NMR) analysis delays) and samples subjected to two variant post-storage conditions in which samples thawed overnight and afterwards (i) were left for 24 h before addition of sodium buffer followed by immediate NMR analysis (buffer delay); and (ii) addition of sodium buffer, then left for 24 h before profiling (NMR delay) (correlations for lipoprotein trait details are given in Appendix A). Spearman’s rank correlation coefficients and 95% confidence intervals (CIs) are listed in Appendix A. C, cholesterol; IDL, intermediate-density lipoprotein; LDL, low-density lipoprotein; HDL, high-density lipoprotein; VLDL, very low-density lipoprotein.

**Figure 8 metabolites-09-00064-f008:**
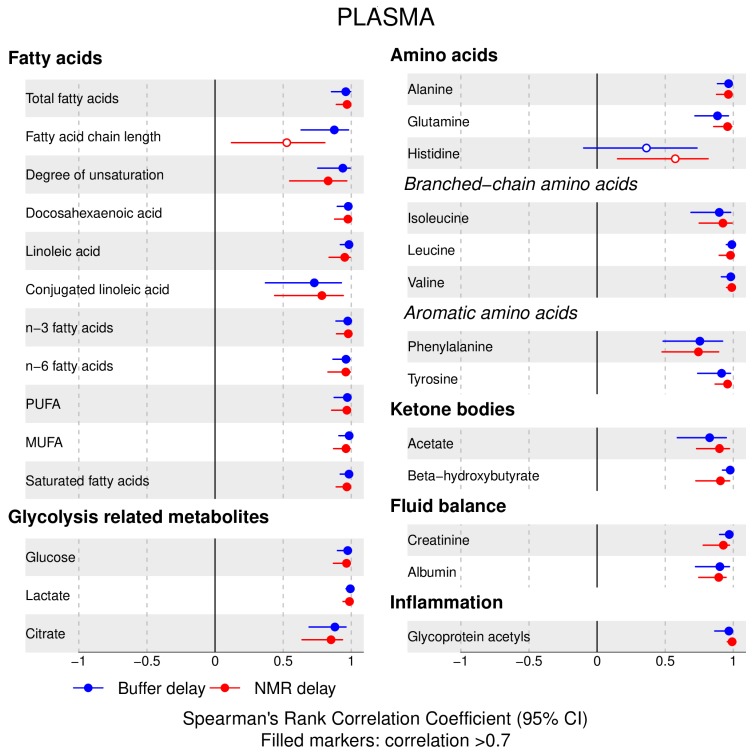
EDTA-plasma, post-storage handling effects (Figure 7 continued): Spearman’s rank correlation coefficients between metabolic trait concentration/value in reference samples (no sample preparation or nuclear magnetic resonance (NMR) analysis delays) and samples subjected to two variant post-storage conditions in which samples thawed overnight and afterwards (i) were left for 24 h before addition of sodium buffer followed by immediate NMR analysis (buffer delay); and (ii) addition of sodium buffer, then left for 24 h before profiling (NMR delay) (correlations for lipoprotein trait details are given in Appendix A). Spearman’s rank correlation coefficients and 95% confidence intervals (CIs) are listed in Appendix A. MUFA, monounsaturated fatty acids; PUFA, polyunsaturated fatty acids.

**Figure 9 metabolites-09-00064-f009:**
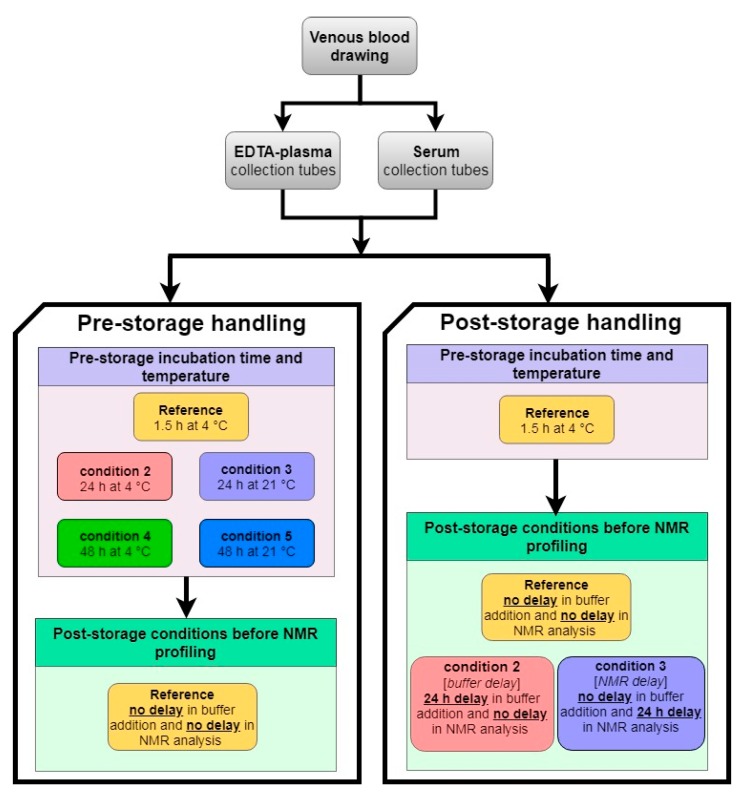
Experimental design diagram: pre- and post-storage handling conditions (Section 4.2 and Section 4.3, respectively). NMR, nuclear magnetic resonance; buffer, sodium phosphate (see Appendix A for more information).

**Table 1 metabolites-09-00064-t001:** Characteristics of study participants (*N* = 37) who contributed to at least one pair of exposure-outcome analysis. Age was collected in categories.

Characteristics	*n*	%
**Female**	29	78
**Age**		
(21–35)	17	46
(36–50)	10	27
(51–65)	10	27
**Ever smoked, No**	15	41
**Time since last alcohol consumption**		
Never Drinks	2	5
<1 Week	16	43
1–4 Weeks	4	11
24 h before blood collection	15	41

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
