# Peer review of "The Effect of Pre-Analytical Conditions on Blood Metabolomics in Epidemiological Studies"

_metabolites, 2019, doi:10.3390/metabo9040064_

Round 1

Reviewer 1 Report

The effect of pre-analytical conditions of blood metabolomics in epidemiological studies

This manuscript examines the effect that pre-storage and post-storage conditions can have on observed serum and EDTA plasma metabolite levels.  Specifically, the authors examine different temperatures and times to initial processing (pre-storage) and the effect of a delay and use of a buffer prior to NMR analysis (post-storage). 

As the authors assert, this is an important question due to the increasingly common use of metabolomics among epidemiologists.  Epidemiologic samples—collected in huge numbers from clinical centers and participant homes—often have sample handling that is less-than-ideal in some respects, and the effects of handling can then distort or attenuate results of research studies.  The experiment used is soundly designed, and the authors use appropriate statistical analyses.

The key flaw of the manuscript, as currently written, is that the authors’ highly quantitative statistical analysis is followed by a mostly qualitative description of the results.  The manuscript needs to more carefully frame and critically evaluate the results to be useful to the field.  Examples and further thoughts below.

The authors assert that “fatty acids, beta-hydroxybutyrate, glycoproteins acetyls, and most lipids and lipoproteins..are robust to incubation temperatures of 4 C or 21 C for up to 48h prior to centrifugation”.  However, the authors provide no definition of “robust” or any context for understanding what the context in which “robust” is to be judged.

Taking just the first section of Table S2, as an example, we see that exactly half of the p-values for extremely large VLDL are below 0.001 for 4 C.  Is this robust?  Why or why not?  What is the threshold?  In a hospital-based case-control of 500 cases and 500 controls in which all samples are handled one way for cases, and another for controls, wouldn’t this effect still be sufficient to cause false positive associations (assuming the true associations are null)?

The authors also argue in the discussion that “91% and 97% of metabolites..had SD differences from the reference below 0.5SD".  Why was 0.5 SD chosen here?  What is the rationale?

In many contexts, 0.5SD could be a large difference.  For cholesterol, for example, many population studies find that 1 SD = ~40 mg/dl, so half of this would be 20 mg/dl.  This is a large effect, particularly when talking about tracking trends over time (such as for the longitudinal cohorts the authors mention).  For reference, JAMA published a paper in 2012 that highlighted a 10 mg/dl change over 22 years in the U.S. (https://jamanetwork.com/journals/jama/fullarticle/1383233). This effect size is technically within the range that could be explained by a 0.5 SD sample handling effect.

My point is that the effect sizes need to be discussed in light of the real problems that sample handling artifacts can cause, with a more sophisticated mathematical reckoning of these problems.  This could include an exploration of how likely sample handling is to cause bias or attenuation or false positives under different circumstances.  At the least, there needs to be a real justification for the criteria for determining what is robust.  Without this, the manuscript seems to be angling for a predetermined conclusion. 

Minor comments

1.  In the literature, other papers refer to what the authors call “Pre-storage” as “Pre-processing”.  If not contraindicated for some other reason, the authors could consider changing their use of the term “pre-storage” to “pre-processing”.

2.  Consider moving Figure S1 to the main body of the manuscript—it is easier to follow than text.  Also, the language in the boxes describing the post-storage conditions is not fully parallel, making it hard to parse what the differences between conditions are.  Consider changing the labels to “delay then buffer”, “no delay, no buffer”, and “buffer then delay”, with a figure legend or footnote that provides the more specific details.

3. In the supplementary tables, please replace 0.0e+00 with a < value (e.g. <0.1e-26).  The original value is confusing; it took me at least 30 seconds to determine whether it was a high or low p-value.

Author Response

Reviewer 1

This manuscript examines the effect that pre-storage and post-storage conditions can have on observed serum and EDTA plasma metabolite levels.  Specifically, the authors examine different temperatures and times to initial processing (pre-storage) and the effect of a delay and use of a buffer prior to NMR analysis (post-storage). As the authors assert, this is an important question due to the increasingly common use of metabolomics among epidemiologists.  Epidemiologic samples—collected in huge numbers from clinical centers and participant homes—often have sample handling that is less-than-ideal in some respects, and the effects of handling can then distort or attenuate results of research studies.  The experiment used is soundly designed, and the authors use appropriate statistical analyses.

Response: Thank you.

The key flaw of the manuscript, as currently written, is that the authors’ highly quantitative statistical analysis is followed by a mostly qualitative description of the results.  The manuscript needs to more carefully frame and critically evaluate the results to be useful to the field.  Examples and further thoughts below.

Response: We agree with the reviewer. For applications of metabolomics in epidemiology, consistency of the metabolic traits’ associations with exposures/outcomes across different pre-analytical conditions is important, more so than the correspondence in mean levels between reference and variant conditions. Overall, association or causal effect estimates will be minimally biased if metabolic traits of participants are ranked correctly according to their metabolite concentration, even if there are differences in mean levels. Therefore, we have re-written the manuscript to focus on the spearman’s rank correlation coefficients results, though we also present differences in mean levels from the multilevel models.

We have:

Figures: moved the multilevel results figures (previously Figure 1-4, now Figures S5-S6) to supplementary material and have added new figures with the spearman’s rank correlation coefficients results (Figures 1-8; and Figures S1-S2).

Tables with the results for the spearman’s rank correlation coefficients have been updated to include 95% confidence intervals computed using bootstrap (now Tables S2-S5)

Re-written abstract, introduction (lines 70-78), results (lines 108-118), discussion and conclusions to focus on the Spearman rank correlation coefficients more so than the differences in mean levels between different conditions. In addition, in the discussion section, we have explored the different ways by which pre-analytical conditions may impact epidemiological studies (lines 326-375).

The authors assert that “fatty acids, beta-hydroxybutyrate, glycoproteins acetyls, and most lipids and lipoproteins are robust to incubation temperatures of 4 C or 21 C for up to 48h prior to centrifugation”.  However, the authors provide no definition of “robust” or any context for understanding what the context in which “robust” is to be judged.

Response:  In clinical chemistry, an analyte is described as unstable if the change in concentration is significantly greater than the maximum allowed inaccuracy according to pre-established guidelines, for example for creatinine, cholesterol and HDL-cholesterol and LDL-cholesterol the maximum percentage change allowed is 6% according to the guidelines of the German Federal Medical Council. Therefore, the criteria for robustness depend on the guideline and the specific metabolite. However, metabolomics quantifies many metabolites that are not yet included in these guidelines. Moreover, these guidelines focus on percentage change which is not always possible to compute for metabolites in which the reference concentration is zero, as is the case for many very-low-density-lipoprotein traits (please see Table S1). Therefore, we do not feel that we can provide a definition of robustness and agree that it was incorrect to use this term as we had. In the revised paper we no longer use the term robust and place more emphasis on the Rank correlation coefficient results and discuss different ways in which any impact of different pre-analytical conditions on metabolite levels might bias subsequent epidemiology results (see response to point 2 above)

Taking just the first section of Table S2, as an example, we see that exactly half of the p-values for extremely large VLDL are below 0.001 for 4 C.  Is this robust?  Why or why not?  What is the threshold?  In a hospital-based case-control of 500 cases and 500 controls in which all samples are handled one way for cases, and another for controls, wouldn’t this effect still be sufficient to cause false positive associations (assuming the true associations are null)?

Response:  We have established a P-value (P-value<0.006) threshold using a well-established method in metabolomics which considers that the metabolic traits are highly correlated. Nonetheless, in accordance with the American Statistical Association recommendations (and others), we describe results based on their magnitude and precision (95% confidence intervals) rather than whether they meet an arbitrary P-value threshold. We have added more information about the P-value threshold method in supplementary material Text S1 and have also added the following to the main manuscript “Statistical Analysis” section:

 “Metabolic traits are highly correlated [42]. Nine principal components, in this study, explain at least 95% the  variance of serum and EDTA-plasma metabolome (please see Text S1), and this number is used to correct conventional P-value thresholds for nominal significance using the Bonferroni method (alpha=0.05/9 = 0.006)[41,43]. Here, we focus on effect size and precision [44,45].”

Regarding cases-control studies, there is a consensus in epidemiology, that both sets of samples should be treated in the same way to avoid biasing analysis, even in contexts outside metabolomic-epidemiology scope. As noted in point 3 above we no longer use the word robust to describe our results but rather discuss potential influences of any effects of pre-analytical conditions on subsequent epidemiological analyses.

The authors also argue in the discussion that “91% and 97% of metabolites..had SD differences from the reference below 0.5SD".  Why was 0.5 SD chosen here?  What is the rationale? In many contexts, 0.5SD could be a large difference.  For cholesterol, for example, many population studies find that 1 SD = ~40 mg/dl, so half of this would be 20 mg/dl.  This is a large effect, particularly when talking about tracking trends over time (such as for the longitudinal cohorts the authors mention).  For reference, JAMA published a paper in 2012 that highlighted a 10 mg/dl change over 22 years in the U.S. (https://jamanetwork.com/journals/jama/fullarticle/1383233). This effect size is technically within the range that could be explained by a 0.5 SD sample handling effect.  My point is that the effect sizes need to be discussed in light of the real problems that sample handling artifacts can cause, with a more sophisticated mathematical reckoning of these problems.  This could include an exploration of how likely sample handling is to cause bias or attenuation or false positives under different circumstances.  At the least, there needs to be a real justification for the criteria for determining what is robust.  Without this, the manuscript seems to be angling for a predetermined conclusion. 

Response:  We agree with the reviewer that the threshold of 0.5SD is rather arbitrary, as is true of p-values and many other thresholds in research. We also agree with this reviewer that for epidemiological association/effect analyses the mean difference in concentrations between reference and variant conditions is likely to be less important than correct ranking. We have therefore re-focused our paper on the later. In addition, in the discussion section, we have explored the different ways of how pre-analytical conditions may impact epidemiological studies (lines 326-375).

Minor comments

In the literature, other papers refer to what the authors call “Pre-storage” as “Pre-processing”.  If not contraindicated for some other reason, the authors could consider changing their use of the term “pre-storage” to “pre-processing”.

Response: Thank you for the opportunity to clarify the use of the term pre-storage instead of pre-processing. In the metabolomics workflow (particularly in untargeted metabolomics) pre-processing is a commonly used term for the several steps the data undergoes after data acquisition (e.g. peak alignment, normalization, etc.) and before statistical analysis. Therefore, we use the term pre-storage to clarify that the phase of the metabolomic workflow we are studying is the phase before long term storage and before instrumental analysis, and not after instrumental analysis (data acquisition). The following has been added to the end of the introduction, of the manuscript, to clarify this term:

 Throughout this manuscript we use the term pre-storage instead of (sample) pre-processing to indicate the phase of the metabolomic workflow that occurs before long-term storage of samples. This was done to avoid confusion with the term (data) pre-processing which is used, in metabolomic studies, to mean the several steps undertaken after metabolomic data acquisition (e.g. peak alignment, normalization etc.).

Consider moving Figure S1 to the main body of the manuscript—it is easier to follow than text.  Also, the language in the boxes describing the post-storage conditions is not fully parallel, making it hard to parse what the differences between conditions are.  Consider changing the labels to “delay then buffer”, “no delay, no buffer”, and “buffer then delay”, with a figure legend or footnote that provides the more specific details.

Response: As suggested, by also reviewer 3, we have updated Figure S1 (now Figure 9 and moved to the main manuscript) to better match the text description in section 2.3 of “Materials and Methods.” The post-storage condition we tested were 24h delay in sample preparation (i.e. buffer addition) and 24h delay in NMR sample analysis. After thawing, sample were subjected to (i) reference conditions: no delays in buffer addition and NMR; (ii) condition 2 (buffer delay): 24h delay in buffer addition and immediate NMR analysis; (iii) condition 3 (NMR delay): no buffer addition delay and 24h delay in NMR analysis. We have also update Tables S6-S9 and Figures S5-S6 to match the new designations: buffer delay instead of thaw-buffer; NMR delay instead of buffer-NMR.

In the supplementary tables, please replace 0.0e+00 with a < value (e.g. <0.1e-26).  The original value is confusing; it took me at least 30 seconds to determine whether it was a high or low p-value.

Response: We thank the reviewer for this comment and have changed the tables accordingly.

Reviewer 2 Report

This manuscript is very well organized, and the experiment is set up is in acceptable manner. It also includes useful comparison against extensive references and has potential to be a good reference paper for anyone who is interested in this type of study. At the current draft, the findings that are unique to this manuscript is not highlighted enough.  Also this manuscript could improve if some recommendation of best practice in sample handling in the research/lab settings and clinical settings.

• The study uses EDTA plasma for the data analysis. There are other suggested anti-coagulants such as Li-Heparin which one may agues more suitable for NMR based metabolomics analysis. 

Could you comment on the reasons why EDTA is chosen? 

Would you expect to see similar metabolic changes under these pre- post condition in the plasma samples what was collected with other anti-coagulants?

Are the samples collected with other anti-coagulants not suitable for the analysis methods referenced here (Nightingale NMR protocol)? If so, please explain why.

• Line283 and line 408 : why were the samples kept in the “DARK”? Is there any evidence to suggest the samples kept in the light will be affected?

• Line 403: was the 45min Serum clotting time kept consistent for all the samples? In our experience, some of the clinical samples have different clotting time. I can imagine working with such a large population samples will have differences in clotting time among the serum samples. would these differences have some impact on the metabolic profile?

• Line 411: “immediately froze and…”  should be “immediately frozen and…”  

• Line 415: what is the average time of the sample on the instrument? The time between when the sample is loaded on to the SampleJet to the end of the NMR data collection.The time sample is on the auto-samplers at 6C is included in the post-storage handling?

• Based of the three references that are cited in through out the paper (Ref 35-37), the NMR data were collected using NOEY and/or CPMG experiments. The Table S10 is indicated as (1H) NMR, Nightingale Health, should include this information to be able to comparable long with other listed studies.

Author Response

Reviewer 2

1. This manuscript is very well organized, and the experiment is set up is in acceptable manner. It also includes useful comparison against extensive references and has potential to be a good reference paper for anyone who is interested in this type of study. At the current draft, the findings that are unique to this manuscript is not highlighted enough.  Also, this manuscript could improve if some recommendation of best practice in sample handling in the research/lab settings and clinical settings.

Response: Thank you for your kind comments. We have re-written the manuscript to focus more on the potential impact of pre-analytical variability on subsequent association or effect analyses in epidemiological studies (see responses to Reviewer 1 above). In addition, to better highlight our contribution to the field we added the following sentence (lines 278-282):

We have explored the rank correlation and differences in mean levels of different pre-analytical conditions on over 151 metabolic traits. The pre-analytical conditions that we have explored reflect realistic situations in large scale epidemiological studies. We believe this is an important contribution to the literature given the increasing use of high throughput metabolomics in stored samples from large epidemiological studies.”

2. The study uses EDTA plasma for the data analysis. There are other suggested anti-coagulants such as Li-Heparin which one may agues more suitable for NMR based metabolomics analysis. Could you comment on the reasons why EDTA is chosen? 

Response: At the time this experiment was conducted, Nightingale Health (targeted Nuclear Magnetic Resonance platform used in this study) could only analyze serum or EDTA-plasma. The reason for the preference of using EDTA being that it produces a small set of well resolved signals in the proton NMR spectrum, that can be accounted for and interfere minimally with quantification of metabolites. Heparin, on the other hand, significantly increases the complexity of the resulting NMR spectrum and interferes with quantification of signals in the 1D proton experiment, which is used in the Nightingale Health platform (except the CPMG experiment, which is not affected by the addition of Heparin). Now however, Nightingale can analyze heparin-plasma and citrate-plasma as well. In addition, EDTA is the most commonly used anti-coagulant for clinical chemistry/biochemistry blood work and therefore majority of cohorts that collect blood will have samples from EDTA-plasma. To the best of our knowledge, there is no consensus on which anti-coagulant is more suited for metabolomics research.

3. Would you expect to see similar metabolic changes under these pre- post condition in the plasma samples what was collected with other anti-coagulants?

Response: Our experimental design is not suited to answer this question as our aim was to study the impact of pre-analytical variability in serum and EDTA-plasma only and not across plasma samples with different anti-coagulants. Nevertheless, literature suggests that different anti-coagulants may yield different plasma metabolomes as they inhibit different proteases therefore pre-analytical variability may impact certain metabolites differently depending on the anti-coagulant used.

4. Are the samples collected with other anti-coagulants not suitable for the analysis methods referenced here (Nightingale NMR protocol)? If so, please explain why.

Response: At the time of this study, only serum or EDTA-plasma could be analyzed by Nightingale Health platform. Now however, it also accommodates heparin-plasma and citrate-plasma. Citrate (metabolite) is not quantifiable for citrate-plasma samples.

5. Line283 and line 408: why were the samples kept in the “DARK”? Is there any evidence to suggest the samples kept in the light will be affected?

Response: There is evidence that some metabolites can be broken down by exposure to natural light (e.g. HDL, apolipoprotein B, bilirubin, beta-carotene, retinol) therefore to minimize potential damage to photosensitive analytes, samples were kept in the dark.

6. Line 403: was the 45min Serum clotting time kept consistent for all the samples? In our experience, some of the clinical samples have different clotting time. I can imagine working with such a large population samples will have differences in clotting time among the serum samples. would these differences have some impact on the metabolic profile?

Response: According to the literature, different clotting-times may yield different serum metabolomes, in our experiment, clotting-time was kept consistent for all samples as we wanted to reduce this variability in our study since assessing its impact was not our aim.

7. Line 411: “immediately froze and…”  should be “immediately frozen and…”  

Response: Thank you, it has been corrected.

8. Line 415: what is the average time of the sample on the instrument?

Response: The average time of the sample on the instrument (i.e. time between a sample being put into the NMR equipment and the next sample being put in) was 5 min 51s.

9. The time between when the sample is loaded on to the SampleJet to the end of the NMR data collection.

Response: Each rack of 96 samples took about 9 h 22 mins to run.

10. The time sample is on the auto-samplers at 6C is included in the post-storage handling?

Response: The delay between loading the samples onto the SampleJet and NMR measurement is not accounted for in the post-storage delay time since the overall delay between loading samples to the SampleJet and NMR analysis is short compared to the post-storage delays, including NMR sample preparation time. Moreover, in each rack of 96 NMR samples, control reference serum samples are used to check for any degradation of metabolites, during this analysis delay, which did not occur in our study.

11. Based on the three references that are cited in throughout the paper (Ref 35-37), the NMR data were collected using NOEY and/or CPMG experiments. The Table S10 is indicated as (1H) NMR, Nightingale Health, should include this information to be able to comparable long with other listed studies.

Response: Thank you for noting this. We have changed Table S10 to include NOESY and CPMG experiments in the Analytical Platform column when referring to Nightingale Health platform.

Reviewer 3 Report

The authors investigated the impact of conditions related to sample storage and processing on NMR serum and plasma metabolite measures. Their study is thoughtfully executed and presented; however, I have a few comments and clarifying questions for the authors to consider.

The description of the post-storage conditions in Figure S1 is confusing. It would be easier to follow if parallel text or flow charts were used. From the text, I gathered that for both conditions 2 and 3 samples were thawed overnight, then either a buffer was added or not and 24 hours later NMR profiling was done (i.e., condition 3: overnight thaw àbufferà24 hoursàNMR vs. condition 2: overnight thawà24 hoursàbufferàNMR vs. ref: overnight thawàbufferàNMR). Is this correct? Please clarify.

Why was 0.006 used as the threshold for significance and what statistical test do these p-values correspond to? I did not see a description of correction for multiple comparisons in the text. If this is the threshold for statistical significance, there are a considerable number of statistically significant differences in addition to glycolysis metabolites and amino acids, which are highlighted by the authors. If these differences are not meaningful owing to small effect size, this needs to be more clearly explained. What is considered a meaningful effect size and how would these differences potentially bias risk estimates in an epidemiological study?

Many metabolomics studies use LC/MS platforms. Would the potential impact on serum and plasma metabolite measures under the conditions tested be similar? Although the authors review the literature in table s10, a succinct discussion of what additional studies, if any, are needed for investigators to successfully pool serum/plasma metabolomics data from different studies is warranted.

Minor comments – the legends for figures 1 and 3 do not need the statement on “pyruvate, glycerol and glycine…” In the discussion (line 205), should “unaffected” be changed to “minimally affected” given that a number of these differences appear statistically significant even if they are not considered meaningful by the authors.

Author Response

Reviewer 3

1. The authors investigated the impact of conditions related to sample storage and processing on NMR serum and plasma metabolite measures. Their study is thoughtfully executed and presented; however, I have a few comments and clarifying questions for the authors to consider.

Response: Thank you.

2. The description of the post-storage conditions in Figure S1 is confusing. It would be easier to follow if parallel text or flow charts were used. From the text, I gathered that for both conditions 2 and 3 samples were thawed overnight, then either a buffer was added or not and 24 hours later NMR profiling was done (i.e., condition 3: overnight thaw àbufferà24 hoursàNMR vs. condition 2: overnight thawà24 hoursàbufferàNMR vs. ref: overnight thawàbufferàNMR). Is this correct? Please clarify.

Response: We have updated Figure S1 (now Figure 9 and moved to the main manuscript) to better match the text description in section 2.3 of “Materials and Methods.” The post-storage condition we tested were 24h delay in sample preparation (i.e. buffer addition) and 24h delay in NMR sample analysis. After thawing, sample were subjected to (i) reference conditions: no delays in buffer addition and NMR; (ii) condition 2 (buffer delay): 24h delay in buffer addition and immediate NMR analysis; (iii) condition 3 (NMR delay): no buffer addition delay and 24h delay in NMR analysis. We have also update Tables S6-S9 and Figures S5-S6 to match the new designations: buffer delay instead of thaw-buffer; NMR delay instead of buffer-NMR.

3. Why was 0.006 used as the threshold for significance and what statistical test do these p-values correspond to? I did not see a description of correction for multiple comparisons in the text. If this is the threshold for statistical significance, there are a considerable number of statistically significant differences in addition to glycolysis metabolites and amino acids, which are highlighted by the authors. If these differences are not meaningful owing to small effect size, this needs to be more clearly explained. What is considered a meaningful effect size and how would these differences potentially bias risk estimates in an epidemiological study?

Response: We have established a P-value (P-value<0.006) threshold using a well-established method in metabolomics which considers that the metabolic traits are highly correlated. We have added more information regarding the P-value threshold method in supplementary material Text S1 and have also added the following to the main manuscript “Statistical Analysis” section:

 “Metabolic traits are highly correlated [42]. Nine principal components, in this study, explain at least 95% the  variance of serum and EDTA-plasma metabolome (please see Text S1), and this number is used to correct conventional P-value thresholds for nominal significance using the Bonferroni method (alpha=0.05/9 = 0.006)[41,43]. Here, we focus on effect size and precision [44,45].”

Nonetheless, in accordance with the American Statistical Association recommendations (and others), we describe results based on their magnitude and precision (95% confidence intervals) rather than whether they meet an arbitrary P-value threshold.

We have re-written the manuscript to focus on the impact of pre-analytical variability on epidemiological study results (see response to reviewer 1 above).

4. Many metabolomics studies use LC/MS platforms. Would the potential impact on serum and plasma metabolite measures under the conditions tested be similar? Although the authors review the literature in table s10, a succinct discussion of what additional studies, if any, are needed for investigators to successfully pool serum/plasma metabolomics data from different studies is warranted.

Response: With quantitative metabolomics platforms, it does not make a fundamental difference whether a metabolic trait is quantified by Nuclear Magnetic Resonance or by alternative analytics such as LC/MS-if each method identifies a particular metabolite, only accuracy and precision of the concentration measured may differ. Therefore, any reliable analytic platform, that can detect, quantify and identify the same panel of metabolic traits as in our study, should be able to detect similar changes in metabolite concentration by pre-analytic condition. To make this clear we have added the following to the discussion (lines 318-322):

“Whilst our study has explored the effects on metabolite concentrations using one quantitative NMR platform, we would anticipate similar results with other metabolomic platforms that measure the same metabolites, as pre-analytic conditions affect specific metabolites (as described above) rather than being related to the type of metabolomic platform.

In the revised manuscript we now discuss the implications of our results when pooling data from different studies (Discussion, lines 326-375).

5. Minor comments – the legends for figures 1 and 3 do not need the statement on “pyruvate, glycerol and glycine…”

Response: These figures are now Figures S5 and S6 in supplementary material and we have removed the that statement as per suggested by this reviewer.

6. In the discussion (line 205), should “unaffected” be changed to “minimally affected” given that a number of these differences appear statistically significant even if they are not considered meaningful by the authors.

Response: We have changed as suggested.

Round 2

Reviewer 1 Report

In this draft, the authors shifted focus to the Spearman correlations, which helped anchor the manuscript.  These correlations are mathematically related to the expected degree of attenuation and the final statistical power of the study.  So, I now see what the effect of sample handling is likely to be under these specific circumstances.  I appreciate this change.

I have two minor requests remaining.  First, the text under the heading "Potential bias in epidemiological studies of pre-analytic conditions" is a bit dense for the lay reader, and could use some simplifying/slight reorganization.

Secondly (and relatedly), though the results are generally reassuring, they assume 1) consistent handling; and 2) no bias in handling with respect to case status of other variable of interest.  Regarding point 1, if sample handling conditions vary to some degree from sample to sample at random, then ranks will not be preserved to the extent indicated by the Spearman correlations here.  Some random variation in sample handling is, however, inevitable, especially if multiple technicians or centers are involved.  This variation would add random error and attenuate results above and beyond what the Spearman correlations in this paper indicate.  

Regarding point 2, if sample handling varies by center, and case ascertainment (likelihood of being a case) also varies by center, then sample handling could bias results, even in a prospective cohort.  Samples from cases also tend to be used or handled more often than those of controls in a prospective study, which could bias results.  This could happen if, for example, eligibility criteria for cases are relaxed to allow 5-15% of cases to have had a prior thaw so that prior GWAS data on these cases could be used.

My specific point here is that the authors are too quick to dismiss potential sources of inconsistency in handling and need to give a more balanced accounting of this.  The specific issues should be better delineated and the exact implications should be readily comprehensible. The structure of the two paragraphs above could be used as a sort of scaffold for this, if needed.

Author Response

Line numbers correspond to the document without tracked changes.

Dear Reviewer 1,

We thank you for your support and for the suggested edits which we have now incorporated. Below we have responded to each of your comments in turn and highlighted where changes have been made to the original manuscript.

We look forward to hearing from you,

With best wishes,

Diana L. Santos Ferreira (on behalf of all authors)

1. In this draft, the authors shifted focus to the Spearman correlations, which helped anchor the manuscript.  These correlations are mathematically related to the expected degree of attenuation and the final statistical power of the study.  So, I now see what the effect of sample handling is likely to be under these specific circumstances.  I appreciate this change.

Response: We thank the reviewer for their support and for the suggested edits which we have now incorporated.

2. I have two minor requests remaining.  First, the text under the heading "Potential bias in epidemiological studies of pre-analytic conditions" is a bit dense for the lay reader and could use some simplifying/slight reorganization.

Response: We have reorganized the section "Potential bias in epidemiological studies of pre-analytic conditions" and have tried to be clearer in the examples given as well as in the terms used.  This section is now organized as follows: In the first paragraph, we discuss how pre-analytical conditions may affect the two different types of epidemiological studies i.e. descriptive and analytical studies and how this was assessed in our study i.e. exploring differences in mean concentration and spearman’s rank correlation, respectively.

Lines 327-343: “Rank correlations are useful for considering any impact of pre-analytical effects on subsequent association or causal effect analyses using metabolomics data (i.e. analytical studies), as bias of these results is likely to be minimal if participants are still ranked similarly with respect to metabolite concentrations. Differences in mean concentrations are important in descriptive studies (…)”

In the second paragraph, we explored different scenarios where pre-analytical variation could introduce random or systematic bias in analytical studies (e.g. [1] multi-center studies; [2] metabolites as outcomes or exposures; [3] prospective, cross-sectionals, case-control studies). We used the examples given by this reviewer to provide a broader view of pre-analytical impact.

Lines 344-383: “In analytical studies, (…) For example, if we were interested in the potential effect of multiple metabolites (exposures) on incident coronary heart disease (CHD) (outcome), systematic bias would occur if the effect of pre-analytic conditions on metabolites differed between those who experienced CHD and those who did not (…). This could occur if samples from cases of CHD are handled more often (e.g. have more freeze-thaw cycles) than those of controls, which may occur in prospective nested case control studies. These cycles may enhance the impact of pre-analytical effects, particularly for labile metabolites. (…) A key thing here is whether BMI (or other exposure) was measured some time before bio-samples are collected (i.e. in prospective studies) or at the same time (cross-sectional studies). It is not plausible that pre-analytical conditions could (retrospectively) influence prior measured BMI and therefore in prospective studies there would be random error with an expectation of bias towards the null. (…) The above discussion and examples refer to single studies with single centres, whereas metabolomic (and other ‘omic analytical studies) are increasingly multi study, something that is important for replication and increasing sample size. Systematic bias could occur in multi-centre studies or collaboration pooling results from a number of different studies, if sample handling and the likelihood of having an outcome (e.g. CHD) or the distribution of an exposure (e.g. BMI) varies by centre.

In the third and last paragraph (lines 384-395), we provide some suggestions to researchers working in metabolomic-epidemiology studies.

3. Secondly (and relatedly), though the results are generally reassuring, they assume 1) consistent handling; and 2) no bias in handling with respect to case status of other variable of interest.  Regarding point 1, if sample handling conditions vary to some degree from sample to sample at random, then ranks will not be preserved to the extent indicated by the Spearman correlations here.  Some random variation in sample handling is, however, inevitable, especially if multiple technicians or centers are involved.  This variation would add random error and attenuate results above and beyond what the Spearman correlations in this paper indicate.  

Response: If sample handling conditions vary at random for the variables tested in our study and in the ranges that we tested (e.g. incubation temperature 4 to 21oC; incubation duration up to 48h, etc.), our results for the spearman’s rank gives an indication of how (and which) metabolite concentrations may change. We set up this experience to test the effect of four variables (likely to vary the most in large-scale epidemiological studies) on metabolite concentration and cannot extrapolate our conclusions to the effect of random variation on other variables. We agree with this reviewer that random variation will attenuate association toward the null and have discussed this in lines 356-361: Random error could also bias results in this example. As researchers at the time of collecting and processing serum/plasma are unlikely to know future risk of CHD and sample processing differences are likely to affect all samples similarly within a study this may be more likely than systematic bias. In this case, effects of the pre-analytical conditions are random with respect to CHD and the expectation with random error is that it biases results towards the null.  

And lines 362-367: “As in the first example, if pre-analytical conditions that result in measurement error in the metabolites are associated with BMI there will be systematic bias, which depending on the extent of measurement error and how strongly, and in which direction, it is related to BMI and confounders, could bias results towards or away from the null to a large or small extent. If the pre-analytical conditions are not related to BMI the expectation would be bias towards the null.

4. Regarding point 2, if sample handling varies by center, and case ascertainment (likelihood of being a case) also varies by center, then sample handling could bias results, even in a prospective cohort.  Samples from cases also tend to be used or handled more often than those of controls in a prospective study, which could bias results.  This could happen if, for example, eligibility criteria for cases are relaxed to allow 5-15% of cases to have had a prior thaw so that prior GWAS data on these cases could be used.

Response: We thank the reviewer for these examples which we have included in the re-writing of the section "Potential bias in epidemiological studies of pre-analytic conditions" and that has helped to give a broader view of the potential impact of pre-analytical variation.

Lines 344-383: “In analytical studies, (…) For example, if we were interested in the potential effect of multiple metabolites (exposures) on incident coronary heart disease (CHD) (outcome), systematic bias would occur if the effect of pre-analytic conditions on metabolites differed between those who experienced CHD and those who did not (…). This could occur if samples from cases of CHD are handled more often (e.g. have more freeze-thaw cycles) than those of controls, which may occur in prospective nested case control studies. These cycles may enhance the impact of pre-analytical effects, particularly for labile metabolites. (…) A key thing here is whether BMI (or other exposure) was measured some time before bio-samples are collected (i.e. in prospective studies) or at the same time (cross-sectional studies). It is not plausible that pre-analytical conditions could (retrospectively) influence prior measured BMI and therefore in prospective studies there would be random error with an expectation of bias towards the null. (…) The above discussion and examples refer to single studies with single centres, whereas metabolomic (and other ‘omic analytical studies) are increasingly multi study, something that is important for replication and increasing sample size. Systematic bias could occur in multi-centre studies or collaboration pooling results from a number of different studies, if sample handling and the likelihood of having an outcome (e.g. CHD) or the distribution of an exposure (e.g. BMI) varies by centre.

5. My specific point here is that the authors are too quick to dismiss potential sources of inconsistency in handling and need to give a more balanced accounting of this.  The specific issues should be better delineated and the exact implications should be readily comprehensible. The structure of the two paragraphs above could be used as a sort of scaffold for this, if needed.

Response: We agree and thank the reviewer for the suggested edits which we incorporated and has made our paper much more balanced.
